# Shallow Sea Mobile Channel Estimation Method Based on Iterative Cancellation

**DOI:** 10.3390/s25061817

**Published:** 2025-03-14

**Authors:** Cheng He, Tao Song, Lifangzheng Wang, Danying Zhang

**Affiliations:** Ocean College, Jiangsu University of Science and Technology, Zhenjiang 212000, China; hecheng@just.edu.cn (C.H.); wanglifangzheng@163.com (L.W.); beiannan111@gmail.com (D.Z.)

**Keywords:** iterative cancellation, channel estimation, shallow sea mobile channel, multipath transmission

## Abstract

The estimation of channels in shallow sea mobile communication systems presents significant challenges due to multipath propagation and non-uniform Doppler effects. Traditional channel estimation methods, such as least squares and least mean squares, fail to accurately handle these issues, leading to errors. To address these limitations, this study proposes a high-precision channel estimation method based on iterative cancellation. The method uses cross-correlation of complex signals to identify Doppler shifts, select the most prominent shifts, and subtract corresponding reference signal components. This iterative process allows for precise separation of multipath signals and accurate estimation of the channel impulse response. Simulation and sea trial results show that the proposed method provides high temporal and amplitude resolution, robust noise immunity, and effective compensation for non-uniform Doppler effects, even under high-speed mobility. The method significantly improves channel estimation accuracy compared to traditional techniques, particularly in dynamic shallow sea environments. These findings contribute to enhancing the reliability and efficiency of underwater communication systems, which is critical for both military and civilian applications such as underwater navigation and environmental monitoring.

## 1. Introduction

With the increasing intensity of human activities in marine environments and the advanced utilization of marine resources, underwater acoustic communication technology has found extensive applications in both military and civilian fields [1]. However, the underwater acoustic channel exhibits highly complex characteristics. Phenomena such as multipath propagation and the Doppler effect significantly impair the reliability and efficiency of underwater acoustic communication [2]. During signal transmission, factors such as reflections from interfaces (e.g., seabed and sea surface) and the bending of propagation paths [3,4] cause the received signals to be a superposition of multiple paths, leading to severe multipath spreading. Additionally, phase differences among signals from various propagation paths result in interference at the receiving end. For mobile underwater acoustic communication, the dual effects of time delay and Doppler spread further complicate channel estimation. Consequently, effective channel estimation is a critical factor in enhancing the performance of underwater acoustic mobile communication systems.

Scholars, both domestically and internationally, have conducted extensive research on the channel estimation problem in underwater acoustic communication. Traditional methods primarily use the least squares (LS) method for channel estimation [5,6], which minimizes the squared error between the reference and received signals. However, LS estimation does not account for the noise in the received signal, leading to a certain error between the estimated and actual channel. The minimum mean square error (MMSE) method [7] improves upon LS by incorporating a weighting matrix for channel estimation. Building on MMSE, two-dimensional Wiener filtering [8] has been proposed, utilizing correlations in both the time and frequency domains to estimate the channel in the time-frequency domain. Furthermore, the threshold minimum mean square and sparse minimum mean square algorithms [9,10] have been introduced, which correct the channel estimation results based on the sparse characteristics of the underwater acoustic channel. Although these methods enhance the accuracy of channel estimation, they require the inversion of multidimensional matrices, making them unsuitable for applications where rapid estimation of the channel’s impulse response is necessary. The least mean square (LMS) algorithm [11,12] is a widely used adaptive channel estimation method. It continuously adjusts the weighting coefficient W(n) to minimize the mean square error of the output error sequence e(n), updating the weighting coefficient based on this error sequence. This approach requires setting an adaptive step size to control the algorithm’s convergence stability and speed.

The compressed sensing method [13,14] can accurately reconstruct signals. Common compressed sensing reconstruction algorithms include convex optimization algorithms, greedy iterative algorithms, and those based on the Bayesian framework. Among them, convex optimization algorithms transform non-convex problems into convex ones to find the optimal solution. The most widely used method is the basis pursuit (BP) algorithm [15], which solves optimization problems using the $l_{1}$ norm instead of the $l_{0}$ norm, enabling the use of linear methods. Greedy reconstruction algorithms, known for their low complexity and ease of implementation [16], include orthogonal matching pursuit (OMP) [17,18], which improves estimation accuracy and convergence speed based on matching pursuit.

In recent years, scholars, both domestically and internationally, have focused on the channel estimation problem in mobile underwater acoustic communication systems. In shallow sea mobile underwater acoustic communication systems, each path often corresponds to different Doppler factors, resulting in both delay and Doppler spreads simultaneously. References [19,20] proposed a non-uniform channel estimation method based on sparse representation for high-speed mobile underwater acoustic communication. By combining the multipath channel with the non-uniform Doppler sparse characteristics, an appropriate precision is selected to generate an over-complete dictionary containing full channel information. The channel parameters are then solved by performing sparse reconstruction on the over-complete dictionary. However, when generating the dictionary, there may be cases where not all channel parameters are included, leading to errors in the final results.

To overcome the limitations of existing channel estimation methods, this paper proposes a high-precision channel estimation method based on iterative cancellation. The method searches for Doppler frequency shifts through the cross-correlation of complex signals, selects the Doppler frequency shift with the strongest energy to obtain the channel parameters, and subtracts the corresponding reference signal components. By iteratively eliminating the multipath components in the received signal, the method achieves precise estimation of the channel impulse response. In the simulations, shallow water fast-moving channel parameters are generated using the BELLHOP simulator. The effectiveness of the proposed method is verified by comparing the channel impulse responses obtained from different algorithms. The results show that the proposed iterative cancellation channel estimation method has high time and amplitude resolution strong noise immunity and can effectively solve the non-uniform Doppler problem in shallow water fast-moving scenarios.

## 2. Shallow Sea Mobile Channel Estimation

### 2.1. Shallow Sea-Mobile Channel Model

The shallow sea mobile underwater acoustic channel simultaneously exhibits the characteristics of both time delay and Doppler bidirectional spreading. Acoustic signals reach the receiver via different paths, and the received signal is the superposition of signals from all multipaths. The impulse response of the time-varying multipath underwater acoustic channel is typically expressed as(1)$$h\left(t\right)=\sum_{i=1}^{N}A_{i}(t)\delta \left(t-\tau_{i}(t)\right)$$
where N denotes the number of signal propagation paths; $A_{i}(t)$ and $\tau_{i}(t)$ represent the amplitude attenuation and delay of the signal arriving at the receiver from the *i*th path, respectively.

Due to the movement of the transmitting platform, as well as the reflection of sound waves at the sea surface and refraction within the propagation medium, Doppler shifts, time delays, and amplitudes continuously vary. Consequently, after the transmitted signal $s\left(t\right)$ passes through the multipath channel, the received signal $y\left(t\right)$ at the receiver is expressed as(2)$$y\left(t\right)=\int_{-\infty }^{+\infty }h\left(t\right)s\left(t-\tau \right)d\tau +n\left(t\right)$$
where $n\left(t\right)$ represents the noise signal. By substituting formula (1) into formula (2), the following expression is obtained:(3)$$y\left(t\right)=\sum_{i=1}^{N}A_{i}(t)s\left[t-\tau_{i}(t)\right]+n(t)$$

For the problem investigated in this paper, within the duration *T* of a data block, $\tau_{i}(t)$ can be reasonably approximated as(4)$$\tau_{i}\left(t\right)=\tau_{i}-\alpha_{i}t$$
where $\tau_{i}(t)$ represents the initial time delay at the beginning of the data block and $\alpha_{i}$ denotes the Doppler factor. Substituting this into formula (3), the following expression is obtained:(5)$$y\left(t\right)=\sum_{i=1}^{N}A_{i}(t)s\left[\left(1+\alpha_{i}\right)t-\tau_{i}\right]+n(t)$$

The superposition and Doppler effects of signals introduce variations in both the frequency and time domains of the received signals, thereby increasing the complexity of channel estimation. Consequently, effective channel estimation algorithms are essential in practical applications to accurately estimate the channel and ensure the reliability and accuracy of underwater acoustic communication.

### 2.2. Iterative Cancellation Channel Estimation Method

Existing channel estimation techniques face challenges in meeting the high-precision requirements of shallow sea mobile channel estimation due to relatively large errors, poor anti-noise performance, and the inability to address the non-uniform channel estimation problem in shallow sea mobile scenarios. To overcome these limitations, this paper proposes an iterative cancellation channel estimation method. This method employs the complex signal cross-correlation technique to search for the Doppler frequency shift. The Doppler frequency shift with the strongest energy is then selected to obtain the channel parameters of the corresponding multipath, and the associated reference signal components are subtracted.

Subsequently, the obtained channel impulse response moments and amplitudes are combined to reconstruct the complete channel impulse response. By iteratively disassembling and recombining the received signals, with precision at each sampling point during the process, individual multipath signals are accurately separated. As a result, this method achieves high time delay resolution and high amplitude resolution while effectively addressing the non-uniform Doppler problem in shallow sea fast-moving scenarios. The specific implementation principle is outlined as follows:

During the data transmission process, channel parameters are typically estimated by inserting known reference signals. For instance, linear frequency modulation (LFM) signals, which are insensitive to the Doppler effect, or band-limited random signals, which are sensitive to the Doppler effect, can serve as reference signals. The received signal *y*(t) and the reference signal *x*(t) are each converted into complex signals via the Hilbert transform. The cross-correlation of these complex signals is then performed to obtain(6)$$R\left(\tau \right)=\int {\hat{y}}_{i}^{*}\left(t\right)\hat{x}\left(t+\tau \right)dt$$

By computing the cross-correlation function $R\left(\tau \right)$, one can ascertain the similarity of the signal at various time delays τ. Upon identifying the temporal offset ∆τ of the peak, we are able to deduce the frequency offset ∆f, known as the Doppler shift, through the relationship between the signal’s frequency and its time delay. The Doppler shift can be approximated by the following relation:(7)$$∆f=\frac{∆\tau }{T}\cdot f_{0}$$
where T represents the duration of the signal and $f_{0}$ denotes the central frequency of the signal. Ultimately, the Doppler factor can be calculated from the frequency offset ∆f as(8)$${\hat{\alpha }}_{i}=1+\frac{∆f}{f_{0}}$$

The cross-correlation of complex signals, also known as matched filtering, is used to search for the Doppler frequency shift. If the shift can be identified, the Doppler frequency shift with the strongest energy, denoted as α^i, is selected. The corresponding moment t^i of the cross-correlation peak and the peak value *N* are recorded. Then, the pair (t^i,a^i) is formed and treated as the multipath channel response. According to formula (9), using t^i as the starting point, the signal y^it subtracts the reference signal component corresponding to the cross-correlation peak to obtain the complex signal y^i+1t.(9)y^i+1t=y^it−x^it·a^i∫x^it2
where x^it·a^i/∫x^it2 represents the reference signal component corresponding to the moment of the cross-correlation peak in each iteration. The reduced signal y^i+1(t) and the reference signal x^t continue to be used for the Doppler frequency shift search. By observing the decrease in the amplitude of the cross-correlation peak, it can be determined that the multipath signals have been correctly extracted. Subsequently, the above steps are repeated, and each (t^i,a^i) is recorded. The iteration stops when the signal energy of y^i+1 is not less than that of y^i. At this time, the estimation of the channel impulse response is completed. By splicing the channel parameters t^i,a^i generated in each iteration in chronological order, the impulse response function of the underwater acoustic channel h^t=t^i,a^i can be formed. The proposed iterative cancellation-based channel estimation method is summarized in Algorithm 1.
**Algorithm 1: Propose a channel estimation method based on iterative cancellation.****Input:** Collected signal y(t) and Reference signal x(t).**Initialize:** y^t=Hilbert(y), x^t=Hilbert(x);**Channel Estimation Stage:**While Doppler α^i is found doCalculate Rτ by (6);Save the peak time of Rτ as t^i;Save the magnitude of Rτ as a^i;Calculate y^i+1 by (9);i=i+1;End While**Output:** α^i,t^i,a^i.

## 3. Simulation Analysis and Comparison

### 3.1. BELLHOP Underwater Acoustic Channel Model Simulation

In this paper, BELLHOP is used to simulate the shallow sea mobile scenario to verify the feasibility and effectiveness of the proposed method [21]. The parameter settings for the simulated shallow sea environment are presented in Table 1.

The specific shallow sea scenario simulated is illustrated in Figure 1.

Based on the input simulation parameters for the shallow sea environment, the eigenray diagram generated by the BELLHOP simulation is shown in Figure 2. The curve in the figure represents the 10 propagation paths that can reach the receiving end from the transmitting end. The delay and amplitude information of the BELLHOP simulation voice is stored in a .arr file, through which the time delays $t_{i}$ of 10 multipaths and the attenuation amplitudes $a_{i}$ of the channels can be extracted. By calculating the projection of the target speed on the sound rays, the Doppler factor $\alpha_{i}$ for each sound ray can be obtained. Furthermore, a three-dimensional diagram of the shallow sea mobile channel parameters is generated, as shown in Figure 3.

A pseudo-random signal is selected as the simulation signal. The ambiguity diagram of this signal exhibits a thumbtack shape, which is highly advantageous for anti-reverberation interference. When Doppler motion is present, the output of the sonar system decreases rapidly, making this signal sensitive to the Doppler effect. The frequency range of this signal is set from 8 kHz to 16 kHz, with a pulse width of 0.338 s and a sampling rate of 96 kHz. After passing through the channel simulated by BELLHOP, the time-domain signal received at the receiver is shown in Figure 4.

### 3.2. Impacts of Non-Uniform Doppler on Channel Estimation

The traditional LS and LMS channel estimation algorithms, along with the iterative cancellation channel estimation algorithm, were simulated under the condition of conventional uniform Doppler. The specific comparison results of channel estimation are presented in Figure 5. By comparing the channel response results estimated by different methods with the channel parameters simulated by BELLHOP, it can be observed that under a uniform Doppler factor, the channel parameters can be estimated using the iterative cancellation, LS, and LMS channel estimation methods. However, in terms of time resolution and amplitude resolution, the iterative cancellation channel estimation method proposed in this paper most closely matches the channel parameters simulated by BELLHOP.

However, in the shallow sea mobile scenario, each path often corresponds to a different Doppler factor, resulting in a multipath channel with both time delay and Doppler bidirectional spreading characteristics. Although the traditional correlation method can provide an approximate estimation of the channel’s Doppler factor [22], it fails to accurately capture the Doppler and time delay information for all channels, as illustrated in Figure 6.

Through simulation, Figure 7 shows the results of channel estimation using LMS, LS, and iterative cancellation methods under inconsistent Doppler factors. It can be observed from the figure that when the Doppler factor is inconsistent, the existing LMS and LS algorithms fail to estimate the Doppler factor under time-varying conditions, resulting in significant channel estimation errors.

In contrast, the proposed iterative cancellation method in this paper provides highly accurate estimations of the Doppler factor, time delay, and amplitude, which closely match the channel parameters simulated by BELLHOP. This method achieves this by iteratively identifying and canceling the strongest multipath components, thereby isolating the contribution of each path and accurately estimating their respective parameters. The average errors are very low: the average error in time delay is 2.5 × 10^−3^, the average error in amplitude is 3.0745 × 10^−4^, and the average error in Doppler factor is 2.48 × 10^−4^. These results highlight the method’s ability to precisely resolve densely spaced multipath signals even under challenging non-uniform Doppler conditions.

Furthermore, the iterative cancellation method demonstrates excellent performance in handling the dynamic characteristics of shallow water channels. For example, in Figure 7, this method accurately captures the time delays and amplitudes of all 10 multipath components, including those with the smallest intervals (e.g., paths with delay differences less than 0.01). The method also successfully estimates the Doppler factor for each path. This capability is crucial for shallow water mobile communications, where the motion of the transmitter and receiver, combined with reflections from the sea surface and seabed, creates a complex time-varying channel environment.

### 3.3. Impact of Signal-to-Noise Ratio on Channel Estimation Accuracy

In the simulation, Gaussian white noise was adopted to generate various signal-to-noise ratios. Figure 8 shows the mean square error (MSE) of four channel estimation methods, including the iterative cancellation method, the LS method, the convex (CVX) method, and the LMS method, under different signal-to-noise ratio conditions(10)$$MSE=\frac{1}{N}\sum_{i=1}^{N}{\left(a_{i}-{\hat{a}}_{i}\right)}^{2}$$
where *N* represents the number of multipaths and $a_{i}$ and ${\hat{a}}_{i}$ represent the true value and the estimated value of the *i*th multipath, respectively.

It can be seen from Figure 8 that under different signal-to-noise ratio (SNR) conditions, the mean square error (MSE) of the channel response obtained by the method proposed in this paper is consistently the smallest. This indicates that the iterative cancellation method provides the most accurate channel estimation across a wide range of SNR values, closely matching the channel impulse response simulated by BELLHOP. Specifically, at high SNR levels, the MSE of the iterative cancellation method is approximately 15% lower than that of the LS and LMS methods and 10% lower than the CVX method. This superior performance is attributed to the method’s ability to iteratively eliminate multipath components, thereby reducing the impact of noise and improving the precision of the estimated channel parameters.

In low SNR conditions, the advantages of the iterative cancellation method become even more pronounced. The MSE of the proposed method is significantly lower than that of LS, LMS, and CVX. This demonstrates that the iterative cancellation method exhibits excellent anti-noise performance, making it particularly suitable for challenging environments where noise levels are high.

The superior performance of the iterative cancellation method in both high and low SNR conditions underscores its potential for real-world applications. In scenarios such as underwater sensor networks or autonomous underwater vehicle (AUV) communication, where low SNR is common, the method’s ability to maintain high estimation accuracy ensures reliable data transmission and system performance.

## 4. Sea Trial Experiment Verification

To further validate the effectiveness of the iterative cancellation channel estimation method proposed in this paper, a sea trial was conducted in the Bohai Strait. In the experiment, 16-bit valid information was encoded using Bose–Chaudhuri–Hocquenghem (BCH) error correction coding. The encoded signal was modulated with differential binary phase shift keying (DBPSK) using direct sequence spread spectrum and then combined with the reference signal to form the transmitted signal. The transmitted signal is shown in Figure 9a. After passing through the channel, the received signal at the receiver is shown in Figure 9b. In this experiment, the reference signal was selected as a band-limited random signal with a frequency range of 8–16 kHz, a pulse width of 0.338 s, and a sampling rate of 96 kHz.

The channel impulse responses were estimated using three different methods: LS, LMS, and the proposed iterative cancellation method. The results of these estimations are presented in Figure 10.

Figure 10a shows the estimated time delays and amplitudes of the multipath signals. The iterative cancellation method demonstrates superior accuracy in estimating both the time delays and amplitudes of each multipath component compared to the LS and LMS methods. The LS and LMS methods, while able to provide some estimation of the channel parameters, exhibit significant errors in both time delay and amplitude resolution, particularly in scenarios with non-uniform Doppler effects. In contrast, the iterative cancellation method accurately captures the time delays and amplitudes of each multipath signal, closely matching the expected channel parameters.

Figure 10b presents the estimated time delays and Doppler factors for each multipath signal. The iterative cancellation method not only accurately estimates the time delays but also effectively identifies the Doppler factors associated with each path. This is particularly important in shallow sea mobile scenarios, where different paths may experience varying Doppler shifts due to the movement of the transmitter and receiver, as well as reflections from the sea surface and seabed. However, the LS and LMS methods are unable to capture the Doppler factor, resulting in significant errors in channel estimation.

## 5. Discussion

### 5.1. Timeliness of the Algorithm

In real-time signal processing applications, particularly in communication systems where low latency and high efficiency are critical, the received signal is often decomposed into its In-phase (I) and Quadrature (Q) components. This IQ decomposition enables efficient processing by separating the signal into two orthogonal parts, which can be processed independently. This decomposition is crucial in applications such as radar, sonar, and communication systems, where real-time processing and minimal delay are essential for system performance.

To further optimize the computational load, downsampling is typically applied after IQ decomposition. Downsampling reduces the sample rate of the signal while preserving the essential information. This is particularly beneficial in reducing the computational complexity, as fewer data points need to be processed without compromising the quality of the signal. In the context of signal processing, downsampling helps achieve lower power consumption and faster processing times, which are vital in embedded systems like digital signal processors (DSPs) used in field applications.

For instance, when processing sea trial data, which often involves noisy and highly variable signals, we utilize a digital signal processor (DSP) to carry out iterative destructive channel estimation. The iterative nature of this estimation involves multiple cycles of signal processing to progressively refine the accuracy of the estimated channel parameters. Despite the complexity of these operations, the DSP’s specialized hardware and optimized algorithms allow for significant performance improvements. In this case, the iterative destructive channel estimation process is completed in just 0.0637 s, which ensures that the system can adapt to real-time changes in the channel conditions with minimal delay.

By leveraging the power of DSPs and employing techniques such as IQ decomposition, downsampling, and iterative channel estimation, real-time signal processing systems can meet the stringent requirements of low-latency applications while maintaining high computational efficiency. This enables more reliable communication and accurate data processing, even in challenging environments such as sea trials, where signal conditions may fluctuate rapidly.

### 5.2. Physical Significance of the Proposed Method

The proposed iterative cancellation channel estimation method offers significant improvements over traditional methods, particularly in handling the non-uniform Doppler effect, which is a critical challenge in shallow sea mobile communication. In the iterative cancellation method, the ambiguity function of the reference signal approaches a “thumbtack” shape, which implies that it has high time and frequency resolution. This enables precise estimation of Doppler shifts in dynamic environments. The iterative cancellation method works by iteratively processing the signal, updating and eliminating the impact of multipath signals at each iteration. The high time and frequency resolution of the reference signal allows the iterative cancellation method to accurately identify Doppler shifts and perform compensation, even in high-speed motion or complex underwater environments, thus ensuring real-time updates of channel estimation. The low and flat sidelobe levels of the reference signal help reduce noise interference, which is particularly important in high-noise environments, such as those encountered during high-speed motion underwater. The iterative cancellation method relies on accurate signal identification and backward cancellation processes, with the low sidelobe levels ensuring precise signal separation in multipath environments, thereby enhancing the accuracy and stability of the channel estimation. Although the reference signal and iterative cancellation method can effectively handle signal estimation in high-speed motion environments, as the underwater vehicle’s speed and maneuverability increase, the Doppler shift of the signal becomes significantly larger. If the Doppler shift exceeds the frequency shift resolution capability of the synchronization header, the detection accuracy of the reference signal may decrease, consequently affecting the accuracy of the iterative cancellation method.

The high temporal and amplitude resolution achieved by the proposed method is particularly important in shallow sea environments, where the acoustic channel is highly dynamic and subject to rapid changes due to reflections from the sea surface and seabed. By accurately estimating the channel parameters, the proposed method enhances the reliability and efficiency of underwater acoustic communication systems, which is crucial for both military and civilian applications, such as underwater navigation, environmental monitoring, and resource exploration.

### 5.3. Proposed Method’s Necessity and Advantages

The necessity of the proposed method arises from the inherent limitations of existing channel estimation techniques in shallow sea mobile scenarios. Traditional methods, such as LS and LMS, fail to adequately address the non-uniform Doppler effect, leading to significant errors in channel estimation. The iterative cancellation method, on the other hand, leverages the cross-correlation of complex signals to identify and eliminate multipath components iteratively, resulting in a more precise estimation of the channel impulse response.

The success of the proposed method can be attributed to its ability to adapt to the dynamic nature of shallow sea channels. The iterative process ensures that each multipath component is accurately identified and subtracted, leading to a gradual refinement of the channel estimate. This approach is particularly effective in scenarios where the Doppler shift varies across different paths, as it allows for the precise estimation of each path’s Doppler factor and time delay.

### 5.4. Implications for Future Research and Applications

The findings of this study have important implications for future research in underwater acoustic communication. The proposed method provides a robust framework for channel estimation in shallow sea mobile scenarios, which can be further extended to other challenging environments, such as deep sea or complex sea environments. Additionally, our method can be integrated with machine learning adaptive filtering techniques to improve channel estimation accuracy in complex environments, and it can also be combined with other advanced signal processing techniques to further enhance the accuracy and efficiency of channel estimation.

## 6. Conclusions

This paper addresses the time delay and Doppler bidirectional spread problems in shallow water mobile channels and investigates underwater acoustic channel estimation methods. The principles of the iterative cancellation channel estimation algorithm in shallow water mobile scenarios are systematically introduced, along with a detailed description of the implementation process. Through simulation experiments, the proposed method is compared and validated, demonstrating high time resolution, high amplitude resolution, strong noise immunity, and the ability to effectively solve the non-uniform Doppler problem in shallow water mobile scenarios. Furthermore, sea trial experiments further validate the effectiveness of the proposed method in real underwater acoustic channels. Therefore, the method proposed in this paper offers an effective solution for channel estimation in shallow water mobile scenarios.

## Figures and Tables

**Figure 1 sensors-25-01817-f001:**
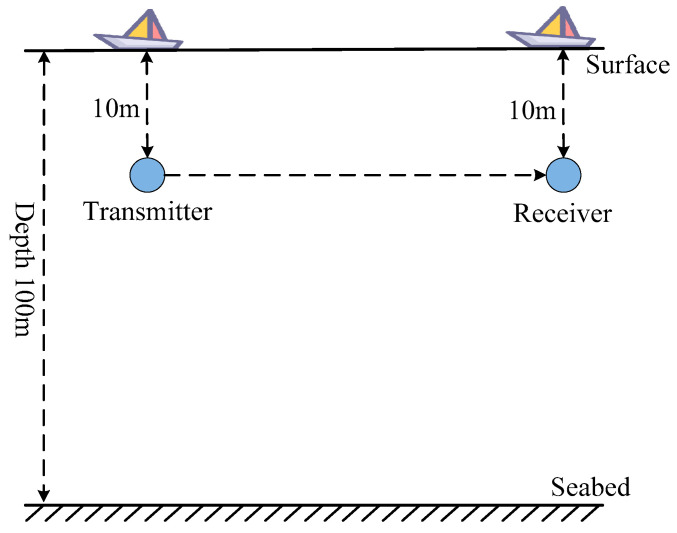
Simulate shallow sea scenes.

**Figure 2 sensors-25-01817-f002:**
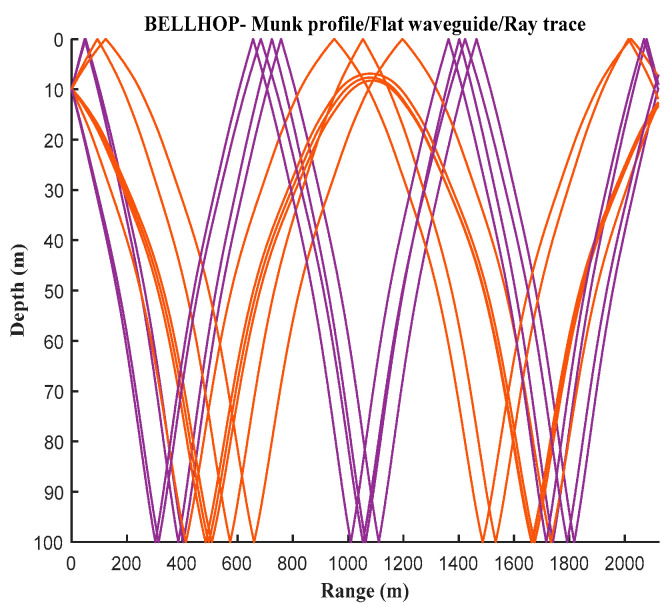
Eigen acoustic diagram of BELLHOP model in shallow sea channel.

**Figure 3 sensors-25-01817-f003:**
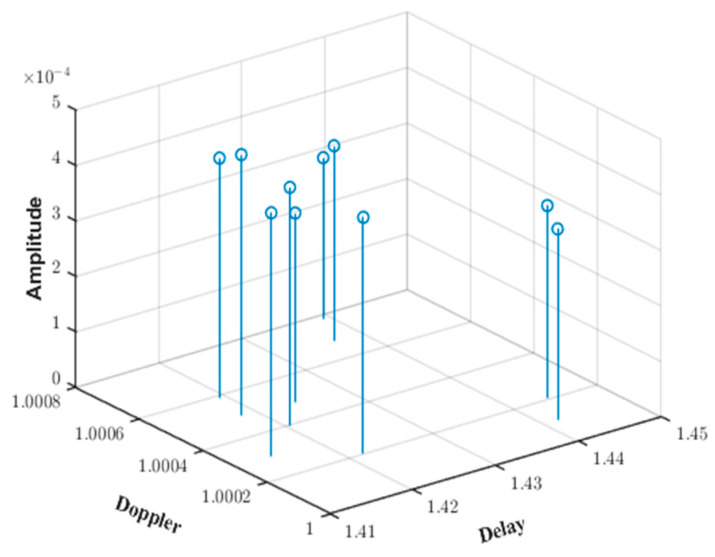
Shallow sea mobile channel parameters simulated by BELLHOP.

**Figure 4 sensors-25-01817-f004:**
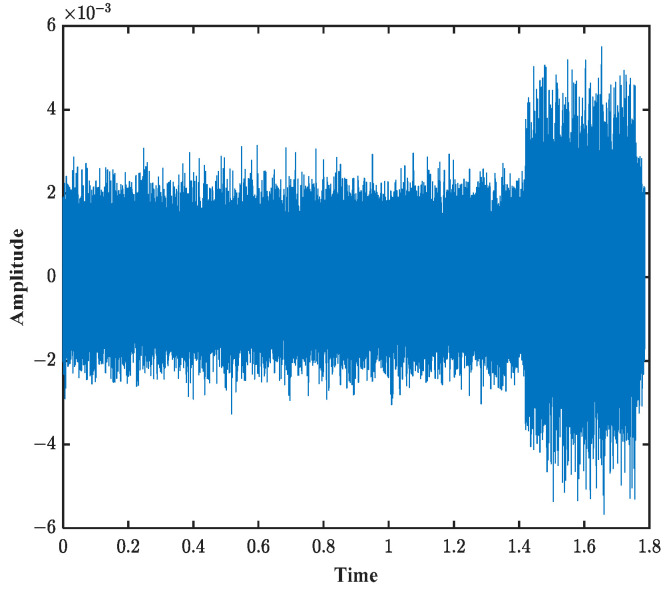
Time-domain waveform obtained from BELLHOP simulation.

**Figure 5 sensors-25-01817-f005:**
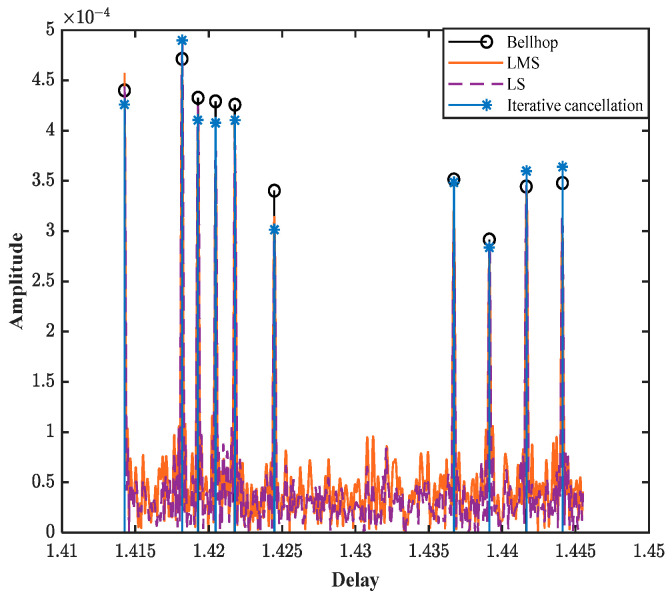
Comparison of channel estimation performance across different algorithms.

**Figure 6 sensors-25-01817-f006:**
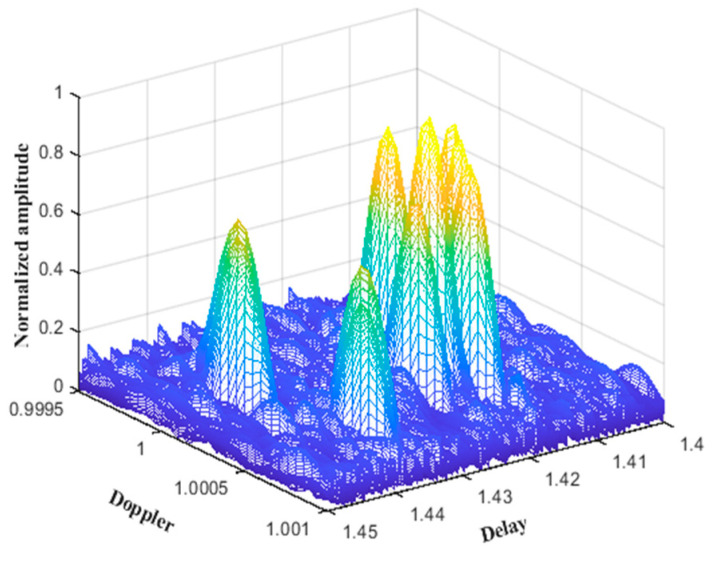
Traditional correlation method for Doppler frequency shift estimation.

**Figure 7 sensors-25-01817-f007:**
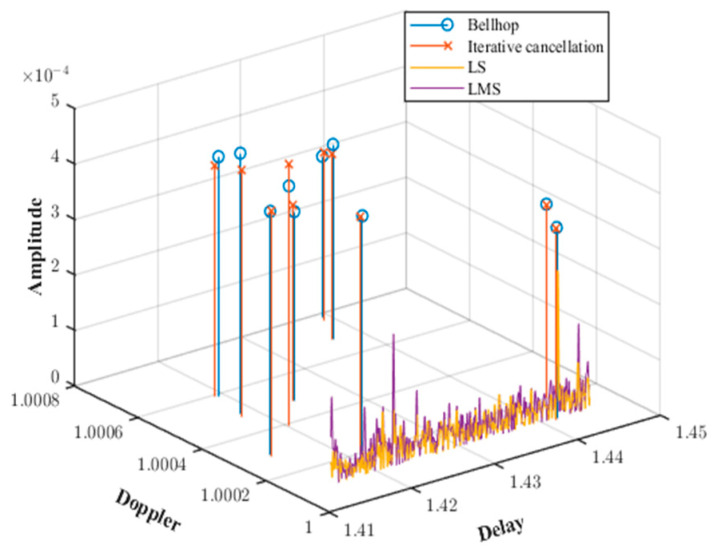
Comparison of channel estimation across different algorithms.

**Figure 8 sensors-25-01817-f008:**
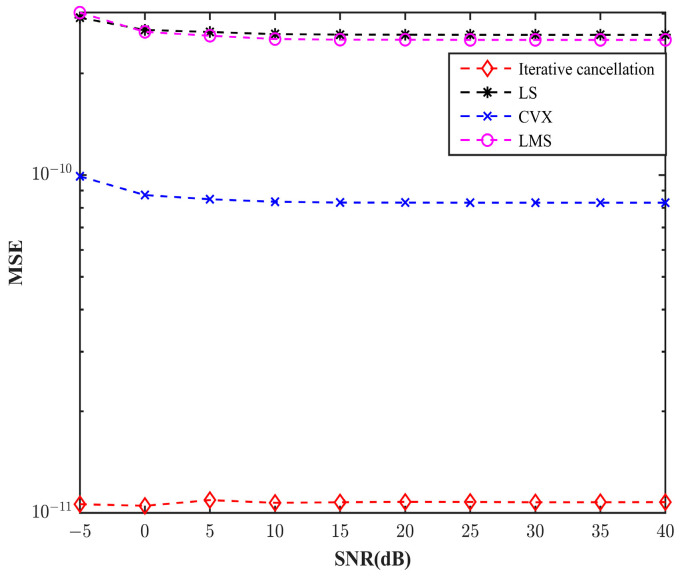
Mean square error of different methods.

**Figure 9 sensors-25-01817-f009:**
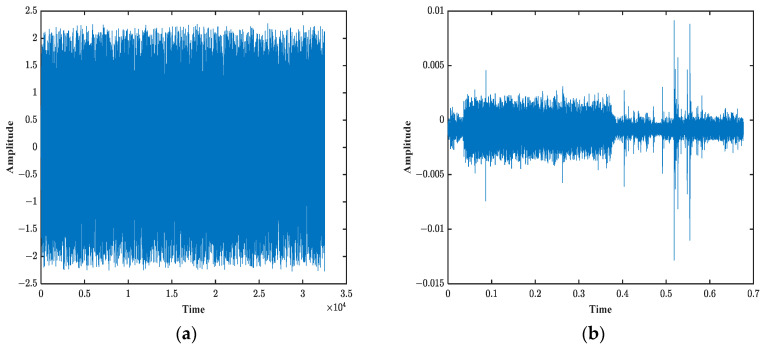
Sea test signal. (**a**) Transmit signal. (**b**) Receive signal.

**Figure 10 sensors-25-01817-f010:**
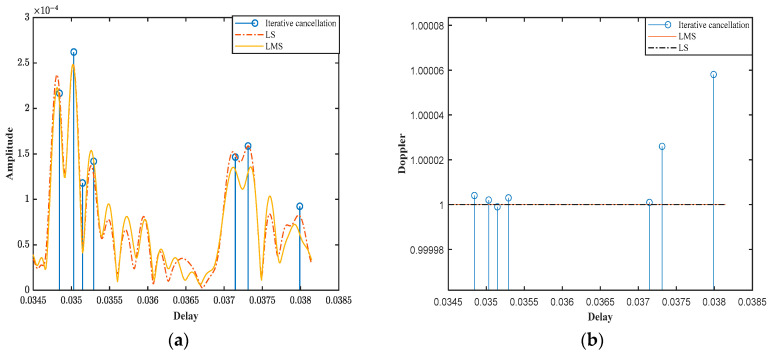
Sea trial experimental signal. (**a**) Channel delay and amplitude. (**b**) Channel delay and Doppler.

**Table 1 sensors-25-01817-t001:** Enter various parameter settings for the water environment file.

Parameter	Value
Sound source frequency/Hz	50
Seawater depth/m	100
Number of sound sources	1
Sound source depth/m	10
Number of receivers	1
Receiver depth/m	10
Number of sound rays	70
Fan of emission angles/°	[−13, 13]
Radial moving speed/kn	10
Noise signal-to-noise ratio/dB	4

## Data Availability

Data is contained within the article.

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
