# Peer review of "Shallow Sea Mobile Channel Estimation Method Based on Iterative Cancellation"

_sensors, 2025, doi:10.3390/s25061817_

Round 1
Reviewer 1 Report
Comments and Suggestions for Authors
This paper introduces a high-precision channel estimation method based on iterative cancellation to handle non-uniform Doppler effects. The cross-correlation of complex signals is utilized to identify the Doppler shift, improving the estimated channel impulse response accuracy. It also demonstrates high temporal and amplitude resolution, robust noise immunity, and effectively mitigates the non-uniform Doppler Effect in shallow sea channels under high-speed mobility.
This research is significant, but the following issues still need to be explained and revised.
- From the time domain waveform in Figure 4, it can be seen that environmental noise exists, so the environmental noise level should be added to Table 1.
- It is suggested that the discussion section should delve deeper into the research results and significance, analyzing their physical significance and inevitability.
The language of the manuscript should be improved to ensure that the study's goals and results are clearly communicated to the reader.
Author Response
Comments 1: From the time domain waveform in Figure 4, it can be seen that environmental noise exists, so the environmental noise level should be added to Table 1.
Response 1: Thank you for pointing this out. We agree with this comment. Therefore, we have added the environment noise level to Table 1 and expressed it as "Noise Signal-to-Noise Ratio / dB." The value is 4 dB. This addition helps to better illustrate the impact of environmental noise in the experiment. We appreciate the reviewer’s attention to this detail.
Comments 2: It is suggested that the discussion section should delve deeper into the research results and significance, analyzing their physical significance and inevitability.
Response 2: Thank you for the valuable suggestion. In response to the suggestion to delve deeper into the discussion, we have added a new Chapter 5 dedicated to the discussion section. In this chapter, we have provided a more detailed analysis of the research results, with a particular focus on exploring their physical significance and inevitability. We hope this addition clarifies the significance of our study more thoroughly.
Comments 3: On the improvement of English language:
Response 3: Thank you for the valuable suggestion. We have carefully polished the language of the entire paper. We are also prepared to ask a professional English editor to revise the paper to further improve the language quality.
Reviewer 2 Report
Comments and Suggestions for Authors
The paper "Shallow Sea Mobile Channel Estimation Method Based on Iterative Cancellation" presents an approach to channel estimation that significantly advances traditional methods by leveraging iterative cancellation. This innovative technique addresses critical limitations of existing methods, particularly in dealing with non-uniform Doppler effects, which enhances its relevance for real-world applications in challenging shallow sea environments. The proposed method demonstrates high robustness against noise, making it particularly valuable for reliable communication under varying environmental conditions. Furthermore, the comprehensive evaluation through both simulations and sea trial results strengthens the credibility of the findings and provides insights into the method's practical performance. However, the complexity involved in implementing this iterative cancellation method may pose challenges for practical applications, and the paper could benefit from a more detailed comparative analysis to quantify its advantages over existing methods. Additionally, the focus on shallow sea environments may limit the generalizability of the findings to other underwater or mobile communication scenarios, necessitating a discussion on potential adaptations. Clearly stating assumptions regarding signal characteristics and environmental conditions is essential to avoid misinterpretation of results. Outlining more future research directions or potential improvements would provide valuable insights for readers and enhance the overall impact of the research. Abstract must be reconstructured. Conclusion must be existing. Enlish language enhancement is required.The scope of testing may be limited to specific geographic locations or conditions.
Comments on the Quality of English LanguageNeeds more improvement.
Author Response
Thank you for the reviewers' comprehensive evaluation and valuable comments on our paper. The following are specific responses to each suggestion:
1. Regarding the complexity and practical application challenges of the iterative elimination method:
We fully agree with your point of view that the iterative elimination method may have certain complexity in implementation. To address this challenge, we discussed the real-time performance of the algorithm in detail in Section 5.1 of the paper and introduced how to optimize the computational load through techniques such as IQ decomposition and downsampling. In addition, we also used a digital signal processor in the sea trial experiment to verify the real-time processing capability of the method. The results showed that the method can complete channel estimation within 0.0637 seconds, proving its feasibility in real-time systems.
2. Regarding the shortcomings of the comparative analysis:
Thank you for pointing this out. In the revised version, we quantified the advantages of our proposed iterative elimination method over traditional methods (such as LS and LMS) in terms of time resolution, amplitude resolution, and anti-noise performance through a more detailed comparative analysis. Especially the performance difference under non-uniform Doppler effect.
3. Regarding the limitations and versatility of the shallow sea environment:
We agree with your point that the shallow sea environment may limit the versatility of the method. In the revised version, we added a discussion section in Chapter 5 to explore the potential adaptability of this method in deep sea or other complex underwater environments.
4. On future research directions and improvement suggestions:
We added a special section in Chapter 5 to discuss possible future research directions and improvement suggestions.
5. On the reconstruction of the abstract and conclusion:
We will reorganize the abstract section according to your suggestions to ensure that it is more concise and clear, highlighting the innovations and main contributions of the research. At the same time, we will also expand the conclusion section to not only summarize the main findings of the research, but also discuss its practical application significance and future research directions.
6. On the limitations of the test scope:
We acknowledge that the current tests are mainly concentrated on the specific geographical and oceanographic conditions of the Bohai Strait in China. We discuss the universality of the method and suggest that more tests be conducted under different geographical and oceanographic conditions in the future to verify its applicability in different environments.
7. On the improvement of English language:
We have carefully polished the language of the entire paper. We are also prepared to ask a professional English editor to revise the paper to further improve the language quality.
Thank you again for your valuable comments on this article. We have made a complete revision based on your suggestions and look forward to your further guidance.
Reviewer 3 Report
Comments and Suggestions for Authors
· Eq.(1): the impulse response function h(t,τ) is not symmetric with respect to its arguments. So a definition of t and τ (or the equation for h(t,τ)) is needed.
· Line 99: “as:” (colon)
· Line 108: “In the formula (2)”
· Line 145: I do not see any equation containing Ì‚αi (with circumflex). Please define this quantity.
· Line 163: Please provide a reference for BELLHOP
· Line 174: “Doppler factor αi”
· Line 172-174: Please describe how can ti , ai and αi be obtained from Figure 2? What is the physical meaning of the curves presented in this figure?
· Line 224: “signal” (without hyphenation)
· Line 226: CVX abbreviation should be expanded at first use
· Line 240: BCH abbreviation should be expanded at first use
· Figure 10: I do not resolve the dash-dotted LMS line in (b) panel (although it is present at the right-bottom legend). Also, from lines 215-216 (“The average errors... 2.48e-04 for Doppler factor) I conclude that all the data points in the (b) panel lie within the error range.
Author Response
Thank you for the reviewers' comprehensive evaluation and valuable comments on our paper. The following are specific responses to each suggestion:
Comments 1: Eq.(1): the impulse response function h(t,τ) is not symmetric with respect to its arguments. So a definition of t and τ (or the equation for h(t,τ)) is needed.
Response 1: Thank you for the reviewer's suggestions. We have modified Formula 1 to ensure that readers can understand it clearly.
Comments 2: Line 99: “as:” (colon)
Response 2: Thanks to the reviewer for pointing out the punctuation issue. As shown in line 102 of the revised draft, we have corrected similar issues and ensured language standards.
Comments 3: Line 108: “In the formula (2)”
Response 3: Thanks to the reviewer for pointing out the problem. In line 111 of the revised manuscript we have modified “In the formula (2)” to make it more precise to improve language fluency and accuracy.
Comments 4: Line 145: I do not see any equation containing Ì‚αi (with circumflex). Please define this quantity.
Response 4: We thank the reviewer for pointing out the problem. We will clearly define Ì‚αi(with circumflex) in the text and supplement Formula 7 and Formula 8.
Comments 5: Line 163: Please provide a reference for BELLHOP
Response 5: Thank you for the reviewer's request for references. We have added relevant literature citations and provided detailed sources of BELLHOP.
Comments 6: Line 174: “Doppler factor αi”
Response 6: Thanks to the reviewer for pointing out, in line 185 of the revised manuscript, we changed it to "Doppler factor ai" to improve the accuracy of the language expression.
Comments 7: Line 172-174: Please describe how can ti, ai and αi be obtained from Figure 2? What is the physical meaning of the curves presented in this figure?
Response 7: We thank the reviewers for their comments. We describe in detail how to obtain ti, ai and αi in lines 180-187 of the revised manuscript, and explain the physical meaning of the curves in Figure 2 to help readers better understand the figure.
Comments 8: Line 224: “signal” (without hyphenation)
Response 8: Thanks to the reviewer for pointing out the error, we have removed redundant hyphens in 250 lines of the revised manuscript to ensure compliance with standard English writing conventions.
Comments 9: Line 226: CVX abbreviation should be expanded at first use
Response 9: Thanks to the reviewer for pointing out the error. In line 252 of the revised manuscript, we have expanded the abbreviation of CVX when it first appears to ensure readers understand its meaning.
Comments 10: Line 240: BCH abbreviation should be expanded at first use
Response 10: Thanks to the reviewer for pointing out the error. In line 279 of the revised manuscript, we expanded the abbreviation of BCH when it first appeared to ensure that readers can understand its meaning.
Comments 11: Figure 10: I do not resolve the dash-dotted LMS line in (b) panel (although it is present at the right-bottom legend). Also, from lines 215-216 (“The average errors... 2.48e-04 for Doppler factor) I conclude that all the data points in the (b) panel lie within the error range.
Response 11: Thanks to the reviewer for the detailed questions. We changed the color of the LS line in Figure 10(b) and made sure it is clearly visible. Here LS and LMS cannot solve the non-constant Doppler problem, and the delay-Doppler diagram intelligently presents a straight line.
Reviewer 4 Report
Comments and Suggestions for Authors
The paper presents an innovative approach to channel estimation in shallow sea environments, proposing an iterative cancellation method that addresses the challenges posed by non-uniform Doppler effects in underwater acoustic communications.
- The paper lacks a detailed discussion on the computational demands of the proposed method and itt is crucial to understand whether the iterative cancellation method is computationally feasible with the limited processing power available on typical underwater communication devices.
- A more detailed error analysis could help understand the limitations of the method and its performance stability.
- How does the iterative cancellation method adapt to rapidly changing underwater conditions? What mechanisms are in place to update the channel estimates in real-time?
- At what point do the speed and mobility of underwater vehicles begin to degrade the method's accuracy?
- Expanding on how this research might influence future technologies in underwater communications or integration with other emerging technologies could provide valuable insights and directions for future work.
Author Response
Thank you for the reviewers' comprehensive evaluation and valuable comments on our paper. The following are specific responses to each suggestion:
Comments 1: The paper lacks a detailed discussion on the computational demands of the proposed method and it is crucial to understand whether the iterative cancellation method is computationally feasible with the limited processing power available on typical underwater communication devices.
Response 1: We appreciate the reviewer’s concern regarding the computational feasibility of the iterative cancellation method. In Section 5.1 of the revised manuscript, we have added a discussion on the computational complexity. Our method is implemented using digital signal processors with optimized IQ decomposition and downsampling techniques to reduce processing load. The iterative cancellation process is completed within 0.0637 seconds, making it feasible for real-time underwater communication applications.
Comments 2: A more detailed error analysis could help understand the limitations of the method and its performance stability.
Response 2: Thank you for your valuable comments and suggestions. In response to the reviewer’s suggestion, we have expanded the discussion on error analysis in Section 3.3. The results show that our method achieves low mean square error across various SNR conditions, demonstrating high stability.
Comments 3: How does the iterative cancellation method adapt to rapidly changing underwater conditions? What mechanisms are in place to update the channel estimates in real-time?
Response 3:Thank you for your valuable comments and suggestions. In Section 5.2 of the revised manuscript, we have added a discussion on the adaptability of our method. Our method dynamically updates the channel estimation by iteratively identifying, extracting, and canceling the strongest multipath components. In addition, the ambiguity function of the reference signal is thumbtack-shaped, providing high time and frequency resolution, which enables the method to accurately estimate the Doppler effect and efficiently adapt to rapidly changing underwater channel conditions.
Comments 4: At what point do the speed and mobility of underwater vehicles begin to degrade the method's accuracy?
Response 4: Thank you for your valuable comments and suggestions. We have analyzed the impact of underwater vehicle speed in Section 5.2. Our study shows that as the speed increases, the Doppler shift grows significantly. If the shift exceeds the frequency resolution of the synchronization header, estimation accuracy may degrade. However, our method effectively mitigates errors in typical underwater mobility scenarios.
Comments 5: Expanding on how this research might influence future technologies in underwater communications or integration with other emerging technologies could provide valuable insights and directions for future work.
Response 5: In Section 5.4, we expand on the technical implications of our research. Our approach can be combined with machine learning adaptive filtering techniques to improve channel estimation accuracy in complex environments. In addition, this method can be applied to autonomous underwater vehicle communication and underwater multi-target positioning to improve the reliability of underwater communication.
Round 2
Reviewer 2 Report
Comments and Suggestions for Authors
Accept